# Morphological Parameters of the Hip Joint and Its Relation to Gender, Joint Side and Age—A CT-Based Study

**DOI:** 10.3390/diagnostics12081774

**Published:** 2022-07-22

**Authors:** Thelonius Hawellek, Marc-Pascal Meier, Mark-Tilman Seitz, Johannes Uhlig, Ali Seif Amir Hosseini, Frank Timo Beil, Wolfgang Lehmann, Jan Hubert

**Affiliations:** 1Department of Trauma Surgery, Orthopedics and Plastic Surgery, University Medical Center Göttingen, 37075 Göttingen, Germany; marc-pascal.meier@med.uni-goettingen.de (M.-P.M.); mark-tilmann.seitz@med.uni-goettingen.de (M.-T.S.); ft.beil@uke.de (F.T.B.); wolfgang.lehmann@med.uni-goettingen.de (W.L.); j.hubert@uke.de (J.H.); 2Department of Diagnostic and Interventional Radiology, University Medical Center Göttingen, 37075 Göttingen, Germany; johannes.uhlig@med.uni-goettingen.de (J.U.); ali.seif@med.uni-goettingen.de (A.S.A.H.); 3Department of Orthopaedics, University Medical Center Hamburg-Eppendorf, 20251 Hamburg, Germany

**Keywords:** hip, morphology, age, gender, joint side

## Abstract

*Background*: Physiological reference values for morphological parameters of the hip (MPH) are of clinical importance for the treatment of painful, degenerated or fractured hip joints, as well as to detect morphological deformities, which could result in early osteoarthritis of the hip. Currently, sufficient data for MPH are lacking. Therefore, it remains unclear if age-dependent alterations in adult hip morphology are physiological and if there are side- and gender-dependent differences. The aim of the study was to analyze MPH according to gender, side and age in a large-scaled cohort by CT scans. *Methods*: A total of 1576 hip joints from 788 patients (female: 257, male: 531; mean age: 58.3 years (±18.9; 18–92 years)) were analyzed by CT. For all hips, acetabular anteversion (AcetAV); lateral centrum edge angle (LCE); acetabular index (AI); femoral neck version (FNV); centrum-collum-diaphyseal angle (CCD); and anterior alpha angle (AαA) were measured. *Results*: The mean values in this cohort were: AcetAV 20.5° (±6.9); LCE 40.8° (±8.8); AI 0.3° (±5.3); FNV 11.0° (±9.8); CCD 129.9° (±7.4); and AαA 41.2° (±7.7). There was a detectable side-specific difference for AcetAV (*p* = 0.001); LCE (*p* < 0.001); CCD (*p* < 0.001); and AαA (*p* < 0.001). All the analyzed parameters showed a significant gender-specific difference, except for AI (*p* = 0.37). There was a significant correlation between age and AcetAV (r = 0.17; *p* < 0.001); LCE (r = 0.39; *p* < 0.001); AI (r = −0.25; *p* < 0.001); CCD (r = −0.15; *p* < 0.001); and AαA (r = 0.09; *p* < 0.001), except FNV (*p* = 0.79). *Conclusions*: There are side-, gender- and age-specific alterations in hip morphology, which have to be considered in treating hip joint pathologies.

## 1. Introduction

Physiological reference values of morphological parameters of the hip (MPH) are essential for treatment decisions in hip joint disease. Morphological alterations in the hip joint can occur on either the acetabular or femoral joint side, respectively, or as combined morphological abnormalities [1,2]. Currently, precursors of osteoarthritis (OA) of the hip are defined as acetabular retroversion (acetabular anteversion: <10°) [3], acetabular dysplasia (lateral centrum edge angle: <20°) [4], and acetabular overcoverage (lateral centrum edge angle: >40° and acetabular index: <0°) [4,5]; coxa retrotorta (femoral neck version: <0°) [6], coxa valga et antetorta (centrum-collum-diaphyseal angle: >135° and femoral neck version: >25°) [7], and coxa vara (centrum-collum-diaphyseal angle: <125°) [3]; and femoral CAM deformity (anterior alpha angle: >55°) [8] (Table 1).

In patients with hip pain and coexisting deformity, hip joint correction surgery should be performed at an early stage [9]. However, there is currently justified criticism when healthy hip joints are classified as too precocious to be pathological and the operation may be carried out too early, as several studies in the past have shown that morphological changes can constantly be detected even in asymptomatic hip joints [10,11,12,13,14]. This controversy underlines that the clinical relevance of the normal physiological diversity for MPH is extremely important to know.

The physiological reference values of MPH were analyzed in the past by varying detection methods such as anatomical examination [15], ultrasounds [16,17], X-ray [3,18] or cross-sectional imaging techniques [3,19,20]. Depending on the specific detection method and study group, the physiological values of MPH in the literature differ significantly.

Currently, radiological examination is used in clinical practice as the standard diagnostic tool for analyzing MPH. However, the hip is a complex 3D joint that makes the use of 2D X-rays for diagnostic examinations inaccurate and inadequate for a comprehensive assessment of hip deformities [10,21]. At worst, these inaccuracies can result in missing and/or mistreating the specific hip joint pathology. Radiological 3D imaging techniques such as MRI or CT scans provide more precise information for the detection of morphological alterations of the hip joint [21]. In this context, it is surprising that to date, radiological reference values for MPH are based in part on studies that were carried out using only X-ray images and/or were partly defined by examinations of selected pathological cohorts, or by studies with a small number of participants [3,22,23].

In addition to corrective surgery, knowledge of MPH is also important for arthroplasty or trauma surgery, in order to reconstruct the degenerated, respectively, fractured hip joint as physiologically as possible, and to enable sufficient postoperative biomechanical function of the hip joint. In these cases, the contralateral hip joint is often used as a preoperative reference. However, it remains unclear whether the symmetry of the right and left hip geometry is consistent [24,25,26]. In addition, it is unclear whether age-related changes in hip morphology appear as physiological in adults and whether there are gender-specific differences that should be taken into account when diagnosing and treating hip pathologies.

The aim of the study was, therefore, to analyze the acetabular and femoral morphological parameters of the hip by gender, side and age in a large, unselected and asymptomatic cohort using CT scans.

## 2. Materials and Methods

### 2.1. Patients

We retrospectively analyzed 1576 CT scans of hip joints that were obtained for reasons other than hip disease from 788 individuals (female: 257, male: 531; mean age: 58.3 years (±18.9; 18–92 years)). The database of analyzed CT scans belonged to polytrauma- and angio-CT-analysis, which were performed between 2007 and 2017 in the University Medical Center Göttingen (UMG). We only included the CT scans of individuals with images of the complete pelvis with an intact left and right femur, without evidence for congenital or developmental structural changes such as dislocation, subluxation or dysplasia; osteolysis, previous surgery or trauma.

The study was approved by the local ethics committee (AN 18/8/17) and is in compliance with the Helsinki Declaration.

### 2.2. Radiological Assessment and Parameters

All subjects had undergone a CT scan (SOMATOM AS+, Siemens Healthineers, Erlangen, Germany). The data were transferred to the PACS system (Picture Archiving and Communication System). The Centricity^TM^ Universal Viewer from GE Healthcare (RA1000, edition 2019, Buckinghamshire, Great Britain) was used as the analysis tool. The CT images had a slice thickness of 0.625–1.5 mm.

All the radiographic parameters in this CT study were manually measured separately for each side of the patient on anonymized imaging files in a standardized manner. The measurements were taken by the same observer (M.-P.M.) under supervision of an experienced radiologist (A.S.). The intra-observer reliability of the measurements of all the parameters was assessed for a subset of 50 subjects by a blinded re-evaluation at 2 weeks after the first measurement and using the same technique. The interobserver reliability was independently assessed by two observers (M.-P.M. and T.H.) for 50 subjects.

The measured MPH included acetabular anteversion (AcetAV); lateral centrum edge angle (LCE); acetabular index (AI); femoral neck version (FNV); centrum-collum-diaphyseal angle (CCD); and anterior alpha angle (AαA). On coronal views, LCE (angle formed by a line parallel to the longitudinal pelvic axis and by the line connecting the center of the femoral head with the lateral edge of the acetabulum [27]); AI (angle formed by a horizontal line and a line through the most medial point of the sclerotic zone of the acetabular roof and the lateral edge of the acetabulum [27]); and CCD (angle by the axis of the femoral neck and the proximal femoral diaphyseal axis [3]) were measured. On axial views, AcetAV (angle between a line drawn between the anterior and posterior acetabular ridges and a reference line drawn perpendicular to a line between the posterior pelvic margins [28]) and AαA (angle formed by the femoral neck axis and a line connecting the center of the femoral head with the point of beginning asphericity [8]) were assessed. FNV was calculated on axial views of the proximal and distal femur by measuring the orientation of the femoral neck in relation to the femoral condyles, according to the technique that was described by Murphy et al. [29]. For the schematic illustration, see Figure 1. If necessary, we manually corrected the pelvic positioning on a sagittal axis for coronal views and on a longitudinal axis for axial views to a neutral orientation to exclude the influence of pelvic malpositioning. The Kellgren–Lawrence classification was used to determine the radiographic osteoarthritis of all the analyzed hip joints [30].

### 2.3. Age Groups

To analyze the differences for MPH according to age, the cohort was divided into four age categories. Age group I (*n* = 390 hip joints) consisted of subjects who were younger than 45 years. In age group II (*n* = 358 hip joints), the subjects had an age between 45 and 62.9 years. Age group III (*n* = 406 hip joints) included all subjects between the age of 63 and 72.9 years, and in age group IV, all subjects (*n* = 422 hip joints) were older than 73 years.

### 2.4. Statistical Analysis

Intra- and interobserver reliabilities were evaluated using intraclass correlation coefficients (ICC). Continuous variables were described as mean values with standard deviation (SD), ranges and percentiles. Categorical variables were provided as absolute numbers and percentage. The patient age groups were stratified according to quartiles. Across subgroups, the continuous variables were compared using the Wilcoxon rank sum test or Kruskal–Wallis test, and the Wilcoxon signed rank test was used for the paired laterality assessment. Categorical variables were compared using the chi-square-test. The Spearman correlation coefficient was used to calculate correlations. Reference values were defined as the 95% reference interval (mean ±1.96 × SD). All statistical analyses were performed using R software (version 3.6.0) and RStudio user interface (version 1.3.959). All provided *p*-values are two-sided. An alpha-level of 0.05 was considered to indicate statistical significance.

## 3. Results

### 3.1. Morphological Parameters of the Hip Joint and OA Grade

Table 2 summarizes the mean values of the acetabular and femoral morphological parameters of the study cohort, according to side and gender. ICC for intra-observer reliability ranged from 0.93 to 0.97 and interobserver reliability from 0.91 to 0.96, indicating excellent reliability. The mean OA grade was 1.51 (±0.8) for the left and 1.46 (±0.8) for the right hip joints. There was no significant difference for the OA grade between the left and right joint side (*p* = 0.2).

#### 3.1.1. Acetabular Anteversion (AcetAV)

The mean AcetAV was 20.5° ± 6.9 (95% reference interval: 7.0–34.0°). There was a significant difference (*p* = 0.001) between the left (19.9° ± 6.5) and right joint side (21.1° ± 7.2). The mean AcetAV for females was 23.3° ± 6.8 and for males 19.2° ± 6.5. There was a significant detectable gender-specific difference for AcetAV (*p* < 0.0001).

#### 3.1.2. Lateral Centrum Edge Angle (LCE)

The mean LCE was 40.8° ± 8.8 (95% reference interval: 23.6–58.0°). The mean LCE was 42.6° ± 8.6 for the left and 38.9° ± 8.6 for the right hip joint. There was a significant detectable side-specific difference for LCE (*p* < 0.001). The mean LCE was 41.8° ± 9.4 for females and 40.3° ± 8.4 for males. There was a significant gender-specific difference for LCE (*p* = 0.003).

#### 3.1.3. Acetabular Index (AI)

The mean AI was 0.3° ± 5.6 (95% reference interval: −10.7–11.8°). There was no significant difference (*p* = 0.58) between the left (0.5° ± 5.4) and right AI (0.1° ± 5.8). The mean AI for females (0.4° ± 6.1) and males (0.2° ± 5.3) also showed no statistical difference (*p* = 0.37).

#### 3.1.4. Femoral Neck Version (FNV)

The mean FNV was 11.0° ± 9.8 (95% reference interval: −8.2–30.2°). There was no difference (*p* = 0.14) between the left (11.4° ± 9.5) and right FNV (10.6 ± 10.0). There was a significant difference (*p* < 0.001) between the mean FNV for females (13.7° ± 10.1) and males (9.7° ± 9.3).

#### 3.1.5. Centrum-Collum-Diaphyseal Angle (CCD)

The mean CCD was 129.9° ± 7.4 (95% reference interval: 115.4–144.4°). There was a significant difference (*p* < 0.001) between the left (131.5° ± 7.2) and right CCD (128.2° ± 7.2). A statistically significant (*p* = 0.006) gender-specific difference between CCD in females (129.1° ± 7.8) and males (130.3° ± 7.2) was evident.

#### 3.1.6. Anterior Alpha Angle (AαA)

The mean AαA was 41.2° ± 7.7 (maximum 95% reference interval: 56.3°). There was a significant difference (*p* < 0.001) between the mean AαA for the left (39.2° ± 7.4) and right hip joint (43.1° ± 7.5). A significant (*p* < 0.0001) gender-specific difference between the mean AαA for females (39.5° ± 7.3) and males (42.0° ± 7.8) could be found.

### 3.2. Association between Age and Alterations of Morphological Parameters of the Hip Joint

The mean of all analyzed parameters of the hip, except for the FNV (*p* = 0.48), showed a significant difference (*p* < 0.001) between the four age groups (Table 3).

In Figure 2, the correlation between the morphological parameters of the hip and age is shown. There was a statistically significant correlation between age and all morphological parameters of the hip, except for the FNV (*p* = 0.79).

### 3.3. Incidence for Pathological Morphological Parameters of the Hip Joint According to Reference Values of the Literature

Table 4 shows the incidence for the pathological morphological parameters of the acetabular or femoral hip joint in our study cohort, according to known reference values in the literature. In 5.5%, the version of the acetabulum was <10°. A dysplastic acetabulum was detected only in 0.3% of all cases. An overcoverage of the acetabulum was found in 24.2% of all the analyzed hip joints. A CAM deformity was detected in 3.4% of cases. A coxa vara was evident in 24.4% of all the analyzed hip joints, while a coxa retrotorta could be found in 11.5% and a coxa valga et antetorta in 1.5%.

## 4. Discussion

Despite having a high clinical relevance for the definition of hip pathologies and surgical reconstruction, there is currently a scarcity of clinical reference data on the morphological parameters of the hip assessed on three-dimensional cross-sectional radiological studies. Therefore, in the current study, we assessed 1576 hip joints of 788 individuals to examine the gender-, side- and age-specific differences of MPH by a CT scan. In addition, the analyzed asymptomatic hip joints were screened for radiological aberrations with respect to the previously published physiological reference values for MPH.

To confirm that there were no radiographic high-grade degenerative changes in the hip joints in the analyzed patient collective, the OA grade was determined according to the Kellgren–Lawrence classification [30]. The radiological health of the hip joints was confirmed with an OA grade mean value Kellgren–Lawrence of <2 for both hip joints.

In our study, a mean AcetAV of 20.5° could be found, which was comparable to the published literature for mean AcetAV ranging from 17° to 23° [22,23,31]. Buller et al. measured a higher mean AcetAV of 26.7° in the general population, but this was detected by a 3D reconstruction analysis of CT scans [32]. Toennies and Heinecke described the physiological range for AcetAV in adults between 15 and 20° [3], which seems to be set too low when comparing to our results and those of the above-mentioned studies for AcetAV [22,23,31,32]. According to our 95% reference interval, the upper limit of the reference range for AcetAV should be corrected to 34°. Like previously described, we found a gender-specific difference for the mean AcetAV, with larger values in females [22,23,31,33]. To date, this is the first study to describe a correlation between age and AcetAV, with larger values in the older population. This might be a result of changes of the posterior wall over age. To our knowledge, until today, there have been no published data for this, and it should be examined by further studies in the future.

In the present study, the mean LCE was 40.8°; in a systemic review, the mean LCE was 31.2° ± 4.9 [13]. Compared to the published range for the physiological acetabular coverage between 25° and 38°, our mean LCE seemed to be high [3]. This discrepancy might be a result of the age of the different analyzed study populations. In the youngest age group in our study (18–44 years), we found a mean LCE of 35.1° ±7.6. In this context, we found a statistically significant correlation between LCE and age when they were assessed as continuous variables. This translated into a clinically relevant difference of LCE = 35.1° (normal acetabular coverage) vs. 43.6° (acetabular overcoverage) in the youngest vs. oldest age quartile in our study. Age-dependent changes in the acetabular morphology are well known in children but have not been described for the adult acetabulum so far. To the best of our knowledge, only the study of Miyasaka et al. [33] analyzed age differences for the LCE in a Japanese study population between individuals <50 (mean LCE 28.5°) and ≧50 years (mean LCE 36.0°) and found a significantly lower mean LCE in the younger age group.

We can support the results of other studies that found higher LCE values in women than in men [9,23,32,34]. In addition, we found joint side differences for the LCE, which is hard to explain and needs further examination. Interestingly, there were no significant differences for gender and side regarding the AI. However, a significant age-dependent change could also be found for the AI.

The physiological reference range for the FNV was published as between 5 and 25° [3,6,15,35]. In line with the literature, the mean FNV of our study was 11.0° ± 9.8. In contrast to these results, Buller et al. published a lower mean FNV of 4.7° ± 7.8° for the left and of 4.5° ± 8.5° for the right hip joint [32]. The differences for the reported mean FNV could be explained by the use of different imaging and measurement techniques [36,37,38]. In our study cohort, there was no side- or age-specific difference for the FNV. However, as previously described, we found a statistically significant gender-specific difference for FNV [37,39,40].

The mean CCD in our study was 129.9°, which is comparable to the results of other studies [15,20,41]. Toennies and Heinecke described a range for the physiological CCD between 125° and 140° [3]. In contrast, Hartel et al. published a lower mean CCD of only 122.2° [35]. We here confirm the results of other authors who reported an inverse correlation between age and the inclination of the CCD [20,35,42].

Our results show a significant side-specific difference for the mean CCD, which is comparable to previously published studies that also found slightly higher CCD values for the left rather than the right proximal femur [40,43]. So far, however, there is no explanation for these differences.

We detected a significant gender-specific difference for the mean CCD between males and females. Interestingly, Jiang et al. also found a higher mean CCD for males than females, but the difference did not reach statistical significance [40]. Hartel et al. previously reported gender-specific CCD differences as well but found a higher mean CCD in women than in men, in contrast to our results [35]. These controversial data show that, to date, a gender-specific effect for the CCD has not yet been proven.

In the present study, the mean AαA was 41.2°, which is consistent with the results of Hack et al. (40.8°) analyzed by MRI [44], and Toogood et al. [15] (45.6°) evaluated by anatomical analyses. In a systemic review, a higher mean AαA (54.1°) in individuals with asymptomatic hips was reported [13]. These divergent results could underline a high variability of the mean AαA in the general population. Further studies must clarify this discrepancy. In line with results by Laborie et al. [45], we found a significantly higher mean AαA in right than in left hip joints.

We detected a gender-specific difference for the AαA with a higher mean AαA for males compared to females, which is comparable to earlier studies [9,15,34,44,46,47]. Like Gollwitzer et al. [46], we found a correlation between age and AαA.

Finally, we analyzed the cohort for morphological aberrations, which are considered to be precursors of hip joint osteoarthritis, as defined by reference values from the literature. We only found <0.5% of individuals with a borderline or dysplastic acetabulum. In contrast, we detected an acetabular overcoverage in approximately 25% of all included cases. These age-dependent findings are in accordance with the results of Nardo et al. [18], who found a prevalence of Pincer deformity in 21% in a cohort of 8151 males aged >65 years. In a systemic review, the overall prevalence of individuals with asymptomatic hips and overcoverage was up to 67% [13]. These results show that the upper limit of the LCE at 40° is possibly too low, especially for older patients. According to our 95% reference interval, the upper limit for the LCE should be extended to 50.0°. In our cohort, a retroversion of the acetabulum was found in 5.5% of all cases. Lerch et al. [1] found a prevalence of 14% for acetabular retroversion, however, in a symptomatic hip joint collective.

Surprisingly, in the present study, a CAM deformity could only be found in 3.4% of cases. Other studies published a prevalence of up to 53% in asymptomatic hip joints [13,44]. We detected a maximum AαA for the 95% reference interval of 56.3°, which corresponds to the upper limit for the AαA of 55°, as defined by Nötzli et al. [8]. We could find a coxa vara in about 25% of all cases. Fischer et al. earlier described that the historical lower limit of 120° for the reference range for the CCD should be corrected [20]. Our results support this recommendation of Fischer et al. to set the lower limit for the reference range of the CCD to 115°. In the present study, the 95% reference range for the CCD was between 115.4° and 144.4°. Surprisingly, in our cohort, a coxa retrotorta was detected in 11.5% of all cases. Lerch et al. published a prevalence of coxa retrotorta of only 5% in a symptomatic hip cohort [1]. The high incidence in our collective remains unclear—in particular, since we used the same methodology as Lerch et al. for the evaluation of the femoral neck version that was described by Murphy et al. [29]. In contrast, the deformity of a coxa valga et antetorta was rarely found in our study population.

There are some limitations to this study. The analyzed cohort may not reflect the average adult population as the data were collected in part from angio-CT scans, including patients with severe primary diseases. In particular, the effect of vascular disease on the MPH has not yet been analyzed; therefore, it remains speculative whether the results are biased. Unfortunately, we do not have any information about the analyzed subjects with regard to clinical symptoms. Neither do we have any information on whether the analyzed subjects exposed their hip joints to enormous strain. In addition, the nutritional status of the subjects is missing, which could also play a role in alterations of the MPH. Finally, the weight and height of the analyzed subjects were not collected and analyzed for the MPH. Despite these limitations, this study reveals solid data that shed new light on MPH.

In conclusion, our large-scale data indicate that the hip joint still undergoes age-dependent morphological changes in adulthood, and that there are physiological gender- and side-specific differences for most MPH. Our results suggest that in the general adult population, the hip deformities of coxa vara and acetabular overcoverage might be overestimated, due to a physiological reference range that was historically set too narrow. These results should be considered in the future for the diagnosis and treatment of hip joint pathologies.

## Figures and Tables

**Figure 1 diagnostics-12-01774-f001:**
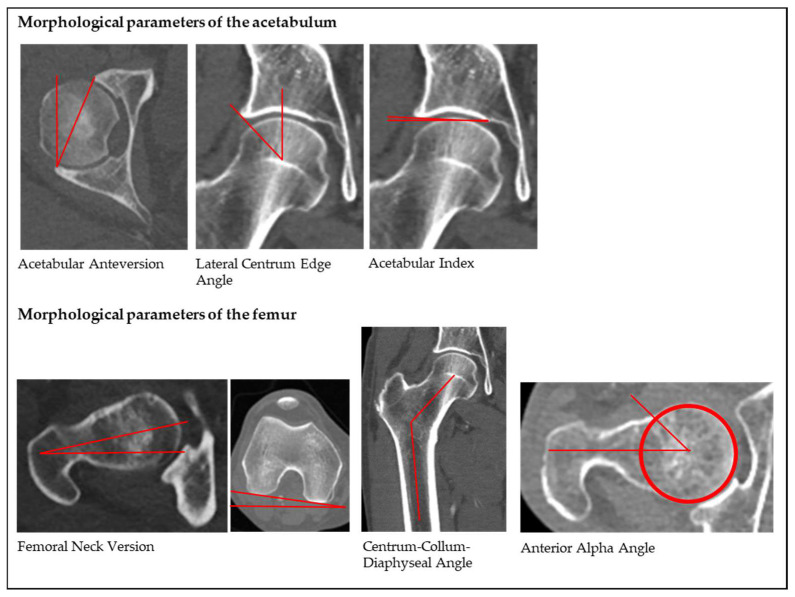
Schematic measurements of the analyzed morphological parameters.

**Figure 2 diagnostics-12-01774-f002:**
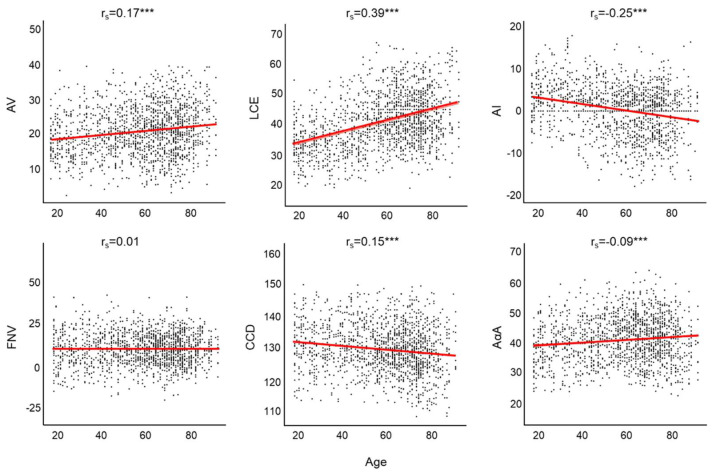
Scatterplot—Correlation analysis between morphological parameters of the acetabulum/femur and age. Abbreviations: AV: acetabular anteversion; LCE: lateral centrum edge angle; AI: acetabular index; FNV: femoral neck version; CCD: centrum-collum-diaphyseal angle; AαA: anterior alpha angle. ***: *p* < 0.001.

**Table 1 diagnostics-12-01774-t001:** Morphological parameters of the hip defined as precursors of osteoarthritis of the hip.

Joint Side	Pathology	Parameter	Study
Acetabular	Acetabular retroversion	AcetAV < 10°	Tönnis D.; Heinecke A. 1999 [3]
	Acetabular dysplasia	LCE < 20°	Tschauner et al., 2002 [2]
	Borderline dysplasia	LCE 20–25°	Lee et al., 2012 [4]
	Acetabular overcoverage	LCE > 40°; AI < 0°	Lee et al., 2012 [4]
Femoral	Femoral CAM deformity	AαA > 55°	Nötzli et al., 2002 [8]
	Coxa vara	CCD < 125°	Tönnis D.; Heinecke A. 1999 [3]
	Coxa retrotorta	FNV < 0°	Hetsroni et al., 2013 [6]
	Coxa valga et antetorta	CCD > 135° + FNV > 25°	Siebenrock et al., 2013 [7]

**Table 2 diagnostics-12-01774-t002:** Mean values of acetabular and femoral morphological parameters, according to side and gender.

	Total (*n* = 1.576)	Left (*n* = 788)	Right (*n* = 788)	*p*-Value	Female (*n* = 514)	Male (*n* = 1062)	*p*-Value
AcetAV	20.5° (±6.9)	19.9° (±6.5)	21.1° (±7.2)	0.001	23.3° (±6.8)	19.2° (±6.5)	<0.0001
LCE	40.8° (±8.8)	42.6° (±8.6)	38.9° (±8.6)	<0.001	41.8° (±9.4)	40.3° (±8.4)	0.003
AI	0.3° (±5.6)	0.5° (±5.4)	0.1° (±5.8)	0.58	0.4° (±6.1)	0.2° (±5.3)	0.37
FNV	11.0° (±9.8)	11.4° (±9.5)	10.6° (±10.0)	0.14	13.7° (±10.1)	9.7° (±9.3)	<0.0001
CCD	129.9° (±7.4)	131.5° (±7.2)	128.2° (±7.2)	<0.001	129.1° (±7.8)	130.3° (±7.2)	0.006
AαA	41.2° (±7.7)	39.2° (±7.4)	43.1° (±7.5)	<0.001	39.5° (±7.3)	42.0° (±7.8)	<0.0001

**Table 3 diagnostics-12-01774-t003:** Mean values of acetabular and femoral morphological parameters for different age groups.

	Total (*n* = 1.576)	Q1: <45 (*n* = 390)	Q2: 45–62.9 (*n* = 358)	Q3: 63–72.9 (*n* = 406)	Q4: >73 (*n* = 422)	*p*-Value
AcetAV	20.5° (±6.9)	19.0° (±6.4)	20.3° (±6.6)	21.1° (±7.2)	21.8° (±7.0)	<0.001
LCE	40.8° (±8.8)	35.1° (±7.6)	41.8° (±8.6)	43.0° (±7.8)	43.6° (±8.5)	<0.001
AI	0.3° (±5.6)	2.6° (±5.4)	0.0° (±5.6)	−1.2° (±5.5)	−0.5° (±5.2)	<0.001
FNV	11.0° (±9.8)	10.8° (±11.6)	11.5° (±9.3)	10.5° (±9.1)	11.1° (±8.7)	0.48
CCD	129.9° (±7.4)	131.4° (±7.4)	130.4° (±6.9)	129.8° (±7.0)	127.9° (±7.8)	<0.001
AαA	41.2° (±7.7)	39.2° (±7.5)	42.8° (±7.3)	42.0° (±8.1)	40.7° (±7.6)	<0.001

**Table 4 diagnostics-12-01774-t004:** Prevalence of morphological aberrations as precursors of osteoarthritis.

Acetabular				
Pathology:	Retroversion [3]	Dysplasia [2]	Borderline [4]	Overcoverage [4]
Parameter:	AcetAV < 10°	LCE < 20°	LCE 20–25°	LCE > 40°; AI < 0
Total in %:	5.5%	0.3%	0.3%	24.2%
Age < 45 years in %:	5.2%	0.5%	0.7%	12.8%
Age > 45 years in %:	5.6%	0.3%	0.1%	28.1%
femoral				
Pathology:	CAM [8]	Coxa vara [3]	Coxa retrotorta [6]	Coxa valga et antetorta [7]
Parameter:	AαA > 55°	CCD < 125°	FNV < 0°	CCD > 135° FNV > 25°
Total in %:	3.4%	24.4%	11.5%	1.5%
Age < 45 years in %:	2.2%	18.7%	16.5%	3.2%
Age > 45 years in %:	3.8%	26.4%	9.7%	0.9%

## Data Availability

All data generated or analysed during this study are included in this published article.

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
