# Peer review of "Morphological Parameters of the Hip Joint and Its Relation to Gender, Joint Side and Age—A CT-Based Study"

_diagnostics, 2022, doi:10.3390/diagnostics12081774_

Round 1
Reviewer 1 Report
Dear Sir/Madame,
thank you very much for asking me to review the paper titled “Morphological parameters of the hip joint and its relation to gender, joint side and age – a CT based study”, submitted for publication in the Diagnostics (ISSN 2075-4418) (Manuscript ID: diagnostics-1834538).
My opinion is that the manuscript is of interest in its field and without apparent errors of fact or logic. The abstract is complete and well written. Content and clarity of the manuscript are satisfactory. The paper is clearly written in proper English. The Authors discuss the paper’s scientific significance. Bibliography is apparently without incorrect self-citation by the authors
In conclusion, my recommendation is that the paper should be accepted as it is
Kindest Regards.
Author Response
Point 1:
Dear Sir/Madame,
thank you very much for asking me to review the paper titled “Morphological parameters of the hip joint and its relation to gender, joint side and age – a CT based study”, submitted for publication in the Diagnostics (ISSN 2075-4418) (Manuscript ID: diagnostics-1834538).
My opinion is that the manuscript is of interest in its field and without apparent errors of fact or logic. The abstract is complete and well written. Content and clarity of the manuscript are satisfactory. The paper is clearly written in proper English. The Authors discuss the paper’s scientific significance. Bibliography is apparently without incorrect self-citation by the authors
In conclusion, my recommendation is that the paper should be accepted as it is
Kindest Regards.
Response 1: Thank you very much for this positive review.
Reviewer 2 Report
The aim of the present article is to analyze morphological parameters of the hip (MPH) according to gender, side and age in a large-scaled cohort by CT scans.
Overall, the paper is nicely written and informative. The introduction provides sufficient background with relevant current literature on the topic. Material and methods are adequately described. The results, data interpretation, and conclusion are solid.
There are only some minor issues, which are reported below, that need to be addressed or fixed.
Introduction, pg. 1: it would be helpful for the reader to provide a table for all the hip parameters which are taken into account as osteoarthritis precursors, such as acetabular retroversion (Acetabular Anteversion: <10°), acetabular dysplasia (Lateral Centrum Edge Angle: <20°), acetabular overcoverage (Lateral Centrum Edge Angle: >40° and Acetabular Index: <0°), coxa retrotorta coxa valga et antetorta and so on…
Paragraph 2.3 and table 2: please revise the number of patients per group, since the sum does not correspond to the total of 788 patients (1576 hip joints).
Discussion: please revise the sentence “Surprisingly in the present study a CAM deformity could only be found in…” since there is something strange reported in the text “Error! Bookmark not defined. Error! Bookmark not defined.”
Author Response
Point 1:
The aim of the present article is to analyze morphological parameters of the hip (MPH) according to gender, side and age in a large-scaled cohort by CT scans.
Overall, the paper is nicely written and informative. The introduction provides sufficient background with relevant current literature on the topic. Material and methods are adequately described. The results, data interpretation, and conclusion are solid.
There are only some minor issues, which are reported below, that need to be addressed or fixed.
.
Response 1: Thank you very much for this positive review.
Point 2:
Introduction, pg. 1: it would be helpful for the reader to provide a table for all the hip parameters which are taken into account as osteoarthritis precursors, such as acetabular retroversion (Acetabular Anteversion: <10°), acetabular dysplasia (Lateral Centrum Edge Angle: <20°), acetabular overcoverage (Lateral Centrum Edge Angle: >40° and Acetabular Index: <0°), coxa retrotorta coxa valga et antetorta and so on…
Response 2: We like to thank for this helpful adivce. To give the reader a better understanding of the subject we added Table 1. “Morphological parameters of the hip defined as precursors of osteoarthritis of the hip” on page 2. The numbering of the existing tables has changed continuously. This has also been changed in the text (page 5 and page 6).
Point 3:
Paragraph 2.3 and table 2: please revise the number of patients per group, since the sum does not correspond to the total of 788 patients (1576 hip joints).
Response 2: We want to thank the reviewer for this important note. We have corrected the number of patients respectively analyzed joints in paragraph 2.3, table 2 and 3. It has been verified that this correction does not affect the results.
Point 3:
Discussion: please revise the sentence “Surprisingly in the present study a CAM deformity could only be found in…” since there is something strange reported in the text “Error! Bookmark not defined. Error! Bookmark not defined.”
Response 3: “Error! Bookmark not defined. Error! Bookmark not defined.” has been deleted. Instead the missing references 13 and 44 were added on page 9.